# Trajectories in muscular strength and physical function among men with and without prostate cancer in the health aging and body composition study

**Alexander R. Lucas**[1,2]*, **Rhonda L. Bitting**[3], **Jason Fanning**[4], **Scott Isom**[5], **W. Jack Rejeski**[4], **Heidi D. Klepin**[3], **Stephen B. Kritchevsky**[6]

**1** Department of Health Behavior and Policy, Virginia Commonwealth University, Richmond, Virginia, United States of America, **2** Division of Cardiology, Department of Internal Medicine, VCU Pauley Heart Center, Richmond, Virginia, United States of America, **3** Department of Internal Medicine, Hematology and Oncology, Wake Forest Baptist Health, Winston-Salem, North Carolina, United States of America, **4** Department of Health & Exercise Science, Wake Forest University, Winston-Salem, North Carolina, United States of America, **5** Department of Biostatistical Sciences, Wake Forest School of Medicine, Winston-Salem, North Carolina, United States of America, **6** Department of Internal Medicine, Sticht Center for Healthy Aging and Alzheimer's Prevention, Wake Forest Baptist Health, Winston-Salem, North Carolina, United States of America

* Alexander.Lucas@vcuhealth.org

**Data Availability Statement:** All relevant data are in Dryad. DOI: (https://doi.org/10.5061/dryad.47d7wm38z).

## Abstract

### Objectives

To examine and compare changes in strength and physical function from pre- to post-diagnosis among men with prostate cancer (PC, [cases]) and matched non-cancer controls identified from the Health, Aging and Body Composition (Health ABC) study.

### Materials and methods

We conducted a longitudinal analysis of 2 strength and 3 physical function-based measures among both cases and controls, identified from a large cohort of community living older adults enrolled in the Health ABC study. We plotted trajectories for each measure and compared cases vs. controls from the point of diagnosis onwards using mixed-effects regression models. For cases only, we examined predictors of poor strength or physical function.

### Results

We identified 117 PC cases and 453 matched non-cancer controls (50% African Americans). At baseline, there were no differences between cases and controls in demographic factors, comorbidities or self-reported physical function; however, cases had slightly better grip strength (44.6 kg vs. 41.0 kg, *p*<0.01), quadriceps strength (360.5 Nm vs. 338.7 Nm, *p* = 0.02) and Health ABC physical performance battery scores (2.4 vs. 2.3, *p* = 0.01). All men experienced similar declines in strength and physical function over an equivalent amount of time. The loss of quad strength was most notable, with losses of nearly two-thirds of baseline strength over approximately 7 years of follow up.

**Funding:** Dr. Lucas's work on this project was partly supported by a National Cancer Institute training grant (R25 CA122061). Dr. Klepin received support from Wake Forest University Claude D. Pepper Older Americans Independence Center (P30-AG21332). Dr. Fanning received support from the National Institute on Aging grants, 5R21AG058249-02, 5P30AG021332-17, and 1R01AG059186-01A1. Scott Isom is supported by National Cancer Institute's Cancer Center Support Grant award number P30CA012197 issued to the Wake Forest Baptist Comprehensive Cancer Center. Dr. Kritchevsky received support from Wake Forest School of Medicine Claude D. Pepper Older Americans Independence Center (P30-AG021332). This research was supported by National Institute on Aging (NIA) Contracts N01-AG-6-2101; N01-AG-6-2103; N01-AG-6-2106; NIA grant R01-AG028050, and NINR grant R01-NR012459. This research was funded in part by the Intramural Research Program of the NIH, National Institute on Aging. The funders had no role in study design, data collection and analysis, decision to publish, or preparation of the manuscript. There was no additional external funding received for this study.

**Competing interests:** The authors have declared that no competing interests exist.

**Abbreviations:** BMI, Body Mass Index; CVD, Cardiovascular Disease; Health ABC, Health Aging and Body Composition; HRQL, Health-related Quality of Life; PC, Prostate Cancer.

## Conclusions

Among both cases and controls, strength and physical function decline with increasing age. The largest declines were seen in lower body strength. Regular assessments should guide lifestyle interventions that can offset age- and treatment-related declines among men with PC.

## Introduction

Prostate cancer (PC) is the most prevalent form of male cancer worldwide, with an estimated 174, 650 new cases being diagnosed in US men in 2018 [1]. Effective treatment and disease management strategies now result in long-term survival [2]. The majority of men remain clinically asymptomatic and if diagnosed with low-risk disease may opt for active surveillance or treatment with surgery and/or radiation. Active surveillance is a promising approach for low risk PC, yet a significant number of men still opt for more aggressive treatments such as surgery and radiation that often result in sexual, bowel and urinary dysfunction [3]. These significant ongoing health-related quality of life (HRQL) challenges may cause men to become depressed, less-active and dis-engaged from activities of daily living (ADL's)[4–6], which could further influence their physical HRQL. Furthermore, disease progression and treatment-related factors can exacerbate already compromised functioning. For example, approximately 50% of men with low-risk disease may eventually receive androgen deprivation therapy (ADT) [7] for recurrent or metastatic disease. ADT leads to well-documented adverse effects, including large declines in strength and physical function [8–11]. What is not well appreciated is how a PC diagnosis in combination with aging-related factors, from pre- to post-diagnosis, can impact physical aspects of HRQL such as strength and physical function. Factors of interest include, comorbidity, body composition, lifestyle behaviors, and psycho-social well-being.

Strength physical function are prognostically important for cancer survivors [12, 13]. In the Health Aging and Body Composition (Health ABC) study, physical function among a mixed group of cancer patients predicted both disability and survival [14]. Several studies have examined physical function cross-sectionally [15] and prospectively [11, 16] following a PC diagnosis, yet few studies have assessed changes from pre- to post-diagnosis or in comparison with that of age-matched non-cancer controls [17]. Alibhai et al., [11] and Pardo et al., [18] examined changes in physical function among patients with PC on different treatments, and Reeve and colleagues compared cases to non-cancer controls [17], however, these studies either examined post-diagnosis changes, which limits our understanding of how pre-diagnosis physical function impacts change post-diagnosis and treatment, or used self-report rather than objective metrics of function [17]. Due to the advanced age of many men diagnosed with PC, it is important to understand how pre-diagnosis levels of strength and physical function influences future risk for decline.

The Health ABC study is a large and well-characterized prospective cohort study of older adults who were healthy at entry into the study, with incident cases of PC and sufficient follow-up data to fully characterize pre- to post-diagnosis changes. A significant strength of the Health ABC study is the racially and ethnically diverse cohort with data on numerous age-related covariates such as comorbidity, body composition, physical activity, and the inclusion of objective muscular strength and physical performance measures. The aims of our study were to: 1) to examine trajectories of objectively measured strength and physical function in PC patients from pre- to post-diagnosis compared to matched non-cancer controls; and 2) to determine whether demographic, treatment or psychosocial factors were associated with

clinically meaningful changes in strength and/or physical function among patients with PC. We hypothesized that patients with PC would have similar trajectories of strength and physical function to non-cancer controls pre-diagnosis, but that following diagnosis, men with PC would experience greater declines than non-cancer controls. Further we hypothesized that among cases, treatment with ADT, more comorbidities, lower levels of physical activity and a greater severity of depressive symptoms would be associated with worse trajectories in strength and physical function than non-cancer controls.

## Materials and methods

The Health ABC Study enrolled 3075 black and white community dwelling older adults. Participants were recruited from a random sample of white Medicare beneficiaries and all age-eligible black residents in designated ZIP code areas in and around Pittsburgh, Pennsylvania, and Memphis, Tennessee, between March 1997 and July 1998. Eligibility criteria for the Health ABC Study were: aged 70 to 79; no difficulty performing activities of daily living, walking one-quarter or a mile, or climbing 10 steps without resting; no reported need of assistive devices for ambulation; no active treatment for cancer in the prior 3 years; no life-threatening illness; and no plans to leave the area for 3 years. All participants provided written informed consent. The Institutional Review Boards at the University of Pittsburgh, the University of Tennessee and the University of California, San Francisco approved the Health ABC protocol. Study participants were contacted every 6 months by telephone or in person and interviewed about health status, hospitalizations, outpatient procedures, and new cancer diagnoses. Information regarding incident cancer diagnoses was also obtained from hospital or clinic records.

### Study sample

For the current analyses, we identified 117 adjudicated PC diagnoses that were confirmed by pathology reports or other medical record information over 10 years of follow-up (study visits 1, 2, 4, 6, 8, 10). An analytic dataset was created to include socio-demographics, strength and physical function outcomes data for all 117 PC cases and 468 matched non-cancer controls (Fig 1). Each participant was assigned an index visit with the participant having had to attempt the 400m walk during the visit and have attempted a 400m walk at a later visit. The Health ABC study collected data on the 400m walk at years 1 (baseline), 2, 4, 6, 8, and 10. For each PC case with an incident diagnosis since Health ABC baseline, we used the last pre-diagnosis visit as an index visit. Frequency matching, weighted by race, was used to randomly assign an index visit for the non-cancer controls at a ratio of 4:1 for a total analytic sample of 585 men.

### Measures

**Muscular strength.**  Muscular strength was assessed with 2 objective measures to capture upper and lower body strength: 1) Isometric grip strength (kg) using a hand-held dynamometer (JAMAR Technologies, Inc., Hatfield, PA); and 2) Quadriceps (quad) strength (Nm) using an isokinetic dynamometer.

**Physical function.**  Physical function was assessed with a further 3 objective measures: 1) a 400m walk test recorded as the proportion of men completing the test at each visit (failure to complete 400m in 15 minutes is indicative of major mobility disability (MMD)[19, 20]); 2) Health ABC physical performance battery (HABCPPB), a combination test (5x timed chair stands, double and single-leg balance tests held for 30 seconds, and narrow walk test for balance and 6m gait speed) designed to measure a wide range of function in the Health ABC cohort of well-functioning older adults. Scores for the HABCPPB are on a scale

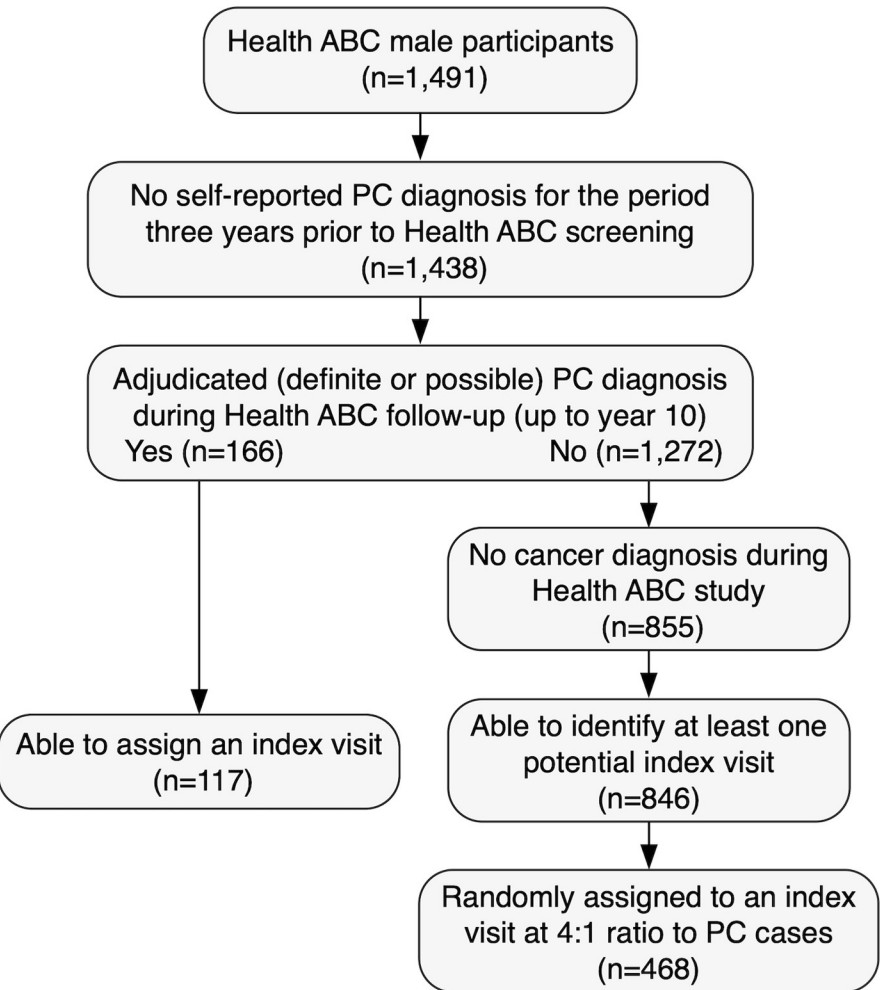

**Fig 1. Participant flow chart.** CONSORT diagram depicting selection of analytic sample.

of 0–4 with a scoring system designed to limit floor and ceiling effects [21]; and 3) 20m gait speed at usual pace over a 20m course assessed on a continuous scale in meters per second (m/s). The correlations between all measures of muscular strength and physical function ranged from 0.24 and 0.37.

**Covariates.** We examined several covariates (as potential confounders of the relationship between PC status and our primary outcomes) and their relationship with muscular strength and physical function among both cases and controls at baseline, index visit and a follow-up visit 2–3 years post-diagnosis. Demographic variables included age at index visit, race, education and marital status. We also considered the presence of multiple comorbidities including diabetes, previous heart attack, hypertension, previous stroke, chronic heart failure (CHF), arthritis, other cancers and total number of comorbidities. Depressive symptoms were measured with the Center for Epidemiologic Studies Depression Scale (CESD) [22]. Body mass index (BMI; kg/m$^2$) was used as an index of obesity status. Self-reported functional status was further assessed by the number of falls in the previous 12 months, perceived capacity to easily walk a quarter mile, lift or carry 10lbs, and having engaged in high intensity exercise over the past 12 months, or past 7 days.

## Statistical analysis

Summary statistics were calculated for both the PC cases and controls at the Health ABC baseline (Table 1), index visit, and at a short-term follow-up visit (2 or 3 years depending on the index year). T-tests were used to compare continuous measures and Chi-square tests were used for categorical measures. To visualize longitudinal data, we plotted the means of each outcome by visit year, centered at the index visit. Nonparametric loess regression was used to plot a smooth curve enabling a better visualization of the effect of time (actual time from diagnosis/index visit) on outcomes. Repeated measures mixed-effects regression analyses were used to test for differences in the effect of time between the two groups and to test whether the slope of the function among cases was different pre- to post-diagnosis. Cubic, quadratic, and linear terms for time were investigated to see which provided the best fit to the data and to verify what was seen in the loess plots. Residual plots were created to verify the assumptions of the model (not shown). To assess the association of relevant covariates with trajectories of upper body strength (grip strength), lower body strength (quad strength) and gait (20m usual pace gait speed), we placed PC cases into 1 of 4 groups for each outcome as follows: group 1—*decreasing*; group 2 –*consistently low*; group 3 –*consistently high*; group 4—*increasing*. Cut offs for determining a meaningful change (±) and the groups men were placed into (*decreasing, consistently low, consistently high or increasing*) are shown in Table 3 and were based on published literature, except in the case of quad strength (where a 1.0 standard deviation (SD) was used). Bivariate association of all covariates was by group (1–4). Loess plots were produced with the ggplot2 package in R 3.4.1 and all other analyses were done using SAS version 9.4 (SAS institute Inc. Cary, NC).

## Results

### Sample characteristics

Briefly, we identified 117 men who reported a diagnosis of PC at a Health ABC follow-up visit and 468 men without cancer, who were randomly chosen from remaining HABC cohort, weighted by race (Fig 1). S1 Table shows sample characteristics at index visit (last visit before diagnosis, including a breakdown of the proportion of patients and matched controls who had their index visit in years 1 (baseline), 2, 4, 6 and 8, which ranged from 9% in year 8 to 33.35% in year 2. The mean age of the PC cases and controls at baseline was similar (74.0 vs. 74.1 years, respectively). Approximately half of cases and controls were black (50.6% vs. 50.4%) with the majority of men having a post-secondary education and being married or partnered. There were no statistically significant differences (all p's >0.05) between cases and controls for demographic factors, comorbidities, self-reported functional capacity or levels of physical activity. There were, however, significant differences in grip strength (mean ± SD: 44.6 ± 8.6kg vs. 41.0 ± 8.7kg; p <.001), quad strength (mean ± SD: 360.5 ± 81.5Nm vs. 338.7 ± 88.5Nm; p <.001), and in HABCPPB scores (mean ± SD: 2.4 ± 0.4 vs. 2.3 ± 0.5 p = 0.01), at baseline. There were no statistically significant differences between the proportion of cases vs. controls completing 400m walk or for 20m gait speed. Four people included in our dataset (all cases) were taking ADT drugs identified in the study medications list (Leuprolide, Goserelin, Buserelin, Nafarelin). Due the small numbers we did not conduct any further analyses using ADT as a sub-population. At index visit, a significantly higher number of controls had a previous heart attack than cases (23.3% vs. 13.7% respectively) while quad strength and HABCPPB scores were no longer significantly different between cases and controls.

**Table 1. Characteristics of the study sample at Health ABC study baseline.**

| | PC Cases (N = 117) Mean (SD) or n (%) | Controls (n = 468) Mean (SD) or n (%) | p-value |
|---|---|---|---|
| *Age at Health ABC baseline* | 74.0 (2.7) | 74.1 (2.8) | 0.61 |
| *Race* | | | 0.97 |
| White | 58 (49.6%) | 231 (49.4%) | |
| Black | 59 (50.4%) | 237 (50.6%) | |
| *Education* | | | 0.08 |
| Less than HS | 32 (27.4%) | 157 (33.5%) | |
| HS grad | 26 (22.2%) | 129 (27.6%) | |
| Postsecondary | 59 (50.4%) | 182 (38.9%) | |
| *Married* | | | 0.10 |
| No | 39 (33.6%) | 116 (25.9%) | |
| Yes | 77 (66.4%) | 332 (74.1%) | |
| *Diabetes* | | | 0.96 |
| No | 94 (80.3%) | 377 (80.6%) | |
| Yes | 23 (19.7%) | 91 (19.4%) | |
| *Heart Attack* | | | 0.10 |
| No | 103 (89.6%) | 386 (83.4%) | |
| Yes | 12 (10.4%) | 77 (16.6%) | |
| *Hypertension/High BP* | | | 0.87 |
| No | 57 (49.6%) | 234 (50.4%) | |
| Yes | 58 (50.4%) | 230 (49.6%) | |
| *Stroke* | | | 0.36 |
| No | 115 (99.1%) | 453 (97.8%) | |
| Yes | 1 (0.9%) | 10 (2.2%) | |
| *CHF* | | | 0.29 |
| No | 113 (98.3%) | 440 (96.3%) | |
| Yes | 2 (1.7%) | 17 (3.7%) | |
| *Arthritis* | | | 0.41 |
| No | 65 (55.6%) | 234 (51.3%) | |
| Yes | 52 (44.4%) | 222 (48.7%) | |
| *Other Cancers* [B] | | | **<0.01** |
| No | 111 (94.9%) | 468 (100.0%) | |
| Yes | 6 (5.1%) | 0 (0.0%) | |
| *Number of comorbidities* [A] | 0.8 (0.8) | 0.9 (0.8) | 0.31 |
| *BMI* | 26.8 (3.6) | 27.0 (4.0) | 0.72 |
| *% Body Fat* | 29.0 (4.3) | 28.9 (5.0) | 0.81 |
| *Lean Body Mass (Kg)* | 54.5 (64.8) | 54.6 (71.9) | 0.87 |
| *CESD* | 3.8 (4.5) | 4.0 (4.5) | 0.69 |
| *Falls in last 12 months* | | | 0.32 |
| No | 99 (85.3%) | 381 (81.4%) | |
| Yes | 17 (14.7%) | 87 (18.6%) | |
| *Easy walking a quarter mile* | | | 0.56 |
| No (Unable to do–Not that easy) | 3 (2.6%) | 17 (3.7%) | |
| Yes (Easy or Somewhat easy) | 112 (97.4%) | 439 (96.3%) | |
| *Easy walking up 10 steps* | | | 0.86 |
| No (Unable to do–Not that easy) | 5 (4.3%) | 18 (3.9%) | |
| Yes (Easy or Somewhat easy) | 111 (95.7%) | 438 (96.1%) | |
| *Easy lifting/carrying 10 pounds* | | | 0.12 |

*(Continued)*

**Table 1.** (Continued)

| | PC Cases (N = 117) Mean (SD) or n (%) | Controls (n = 468) Mean (SD) or n (%) | p-value |
|---|---|---|---|
| *No (Unable to do–Not that easy)* | 1 (0.9%) | 17 (3.6%) | |
| *Yes (Easy or Somewhat easy)* | 116 (99.1%) | 451 (96.4%) | |
| **Past 12 months high intensity exercise** | | | 0.82 |
| *No* | 86 (73.5%) | 348 (74.5%) | |
| *Yes* | 31 (26.5%) | 119 (25.5%) | |
| **Past 7 days high intensity exercise** | | | 0.35 |
| *No* | 100 (85.5%) | 382 (81.8%) | |
| *Yes* | 17 (14.5%) | 85 (18.2%) | |
| **Hours per week watching TV** | | | 0.19 |
| *0—<7* | 17 (14.5%) | 84 (18.1%) | |
| *7—<14* | 36 (30.8%) | 100 (21.5%) | |
| *14—<21* | 26 (22.2%) | 120 (25.8%) | |
| *21 +* | 38 (32.5%) | 161 (34.6%) | |
| **Completed 400m walk** | | | 0.33 |
| *No* | 20 (17.1%) | 99 (21.2%) | |
| *Yes* | 97 (82.9%) | 369 (78.8%) | |
| **20m gait speed (m/sec)** | 1.3 (0.2) | 1.3 (0.2) | 0.08 |
| **HABCPPB** | 2.4 (0.4) | 2.3 (0.5) | **0.01** |
| **Grip Strength** [C] | 44.6 (8.6) | 41.0 (8.7) | **<0.01** |
| **Isokinetic Quad Strength** | 360.5 (81.5) | 338.7 (88.5) | **0.02** |

PC, Prostate Cancer; CHF, Chronic Heart Failure; BMI, Body Mass Index; CESD, Center for Epidemiologic Studies Depression Scale; HABCPPB, Health Aging and Body Composition Physical Performance Battery;

[A] diabetes, heart attack, hypertension/high blood pressure, stroke, CHF;

[B] by design of our sample there are no cancer in control group,

[C] max of two hands.

**Bold** figures indicate statistical significance.

## Trajectories in strength and physical function

Fig 2 (panel A-E) shows the group-based trajectories of change in strength and physical function, from pre- to post-diagnosis. Plot A shows a different pattern in the trajectory of grip strength for cases compared with controls. In controls, grip strength declines by approximately 16% in a steady linear fashion. Among cases, grip strength increases slightly in the years pre-diagnosis but then declines from 2 years prior to diagnosis to 9 years post-diagnosis. Plot B shows that quad strength follows a different pattern to grip strength scores. Cases and controls show similar trends, declining in quad strength at a slower rate initially, but then a much faster rate from about 3 years post-diagnosis, losing approximately two-thirds of their strength over the last 6–7 years of follow-up. Plot C shows the percentage of Health ABC participants completing the 400m walk test. Among both cases and controls, there was a decrease in the proportion of men completing the 400m at follow-up visits, however, the rate of decline in cases and controls appeared to differ depending on the timeframe (pre- or post-diagnosis). Plot D indicates a similar decline in HABCPPB performance scores among both cases and controls over approximately 16 years of total follow-up. Plot E, in a similar fashion to plot D, shows gait speed over 20m follows a pattern of steady, but gentle decline in both cases and controls. Across measures, there are no discernable changes in pattern associated with the point of diagnosis.

**Table 2. Repeated measures mixed-effects regression models of change in physical function by prostate cancer status.**

| Model | | beta | se | p-value |
|---|---|---|---|---|
| **Grip strength** | | | | |
| | Time (years pre/post diagnosis) | -0.70 | 0.03 | **<0.01** |
| | Prostate cancer status (cases vs control) | 1.90 | 0.81 | **0.02** |
| | Time since diagnosis (cancer only) | -0.24 | 0.10 | **0.02** |
| **Quad strength** | | | | |
| | Time (years pre/post diagnosis) | -29.18 | 0.51 | **<0.01** |
| | Time$^2$ (years pre/post diagnosis) | -0.92 | 0.09 | **<0.01** |
| | Prostate cancer status (cases vs controls) | -6.39 | 10.01 | 0.52 |
| | Time since diagnosis (cases only) | -3.13 | 1.65 | **0.06** |
| **Completed 400m walk** | | | | |
| | Time (years pre/post diagnosis) | -0.17 | 0.01 | **<0.01** |
| | Prostate cancer status (cases vs controls) | -0.10 | 0.17 | 0.56 |
| | Time since diagnosis (cases only) | 0.09 | 0.05 | **0.04** |
| **HABCPPB** | | | | |
| | Time (years pre/post diagnosis) | -0.08 | 0.004 | **<0.01** |
| | Time$^2$ (years pre/post diagnosis) | -0.002 | 0.0006 | **0.01** |
| | Prostate cancer status (cases vs controls) | 0.06 | 0.06 | 0.29 |
| | Time since diagnosis (cases only) | 0.02 | 0.01 | 0.16 |
| **20m gait speed** | | | | |
| | Time (years pre/post diagnosis) | -0.04 | 0.001 | **<0.01** |
| | Time$^2$ (years pre/post diagnosis) | 0.0005 | 0.0002 | **0.01** |
| | Prostate cancer status (cases vs controls) | -0.003 | 0.02 | 0.89 |
| | Time since diagnosis (cases only) | 0.008 | 0.004 | **0.02** |

HABCPPB, Health Aging and Body Composition Physical Performance Battery. **Bold** figures indicate statistical significance

Table 2 shows the results of repeated measures models for each outcome. All models include a linear effect for time. For quad strength, HABCPPB, and gait speed there was a significant quadratic effect for time, a comparison of functional trajectory in cases vs controls, and effect of time since diagnosis in PC cases. Grip strength declined significantly over time in both groups ($p<0.01$), though less quickly in cancer cases compared to controls ($p = 0.02$). Among cases, from the point of diagnosis onwards grip strength declined more rapidly than before diagnosis ($p = 0.02$). Quad strength declined significantly over time among both cases and controls ($p<0.01$) and at the same rate ($p = 0.52$). For the proportion of cases and controls completing the 400m walk test, the effects of both time ($p<0.01$) and time since diagnosis ($p<0.04$) were significant, though in opposite directions. In general, fewer cases and controls completed

**Table 3. Change in physical function among PC cases at 3–4 years post-diagnosis.**

| | Median baseline value (to determine high/low) | Criteria for Change | Follow-up Status | | | |
|---|---|---|---|---|---|---|
| | | | **Decreasing** | **Consistently Low** | **Consistently High** | **Increasing** |
| **Grip strength (Kg)** | 42 | ±6.5[A] | 18 (24%) | 29 (39%) | 26 (35%) | 2 (3%) |
| **Quad strength (Nm)** | 322 | ±100[B] | 33 (46%) | 16 (23%) | 22 (31%) | 0 |
| **20m gait speed (m/s)** | 1.2 | ±0.05[A] | 50 (65%) | 10 (13%) | 10 (13%) | 7 (9%) |

[A] = reported in the literature,

[B] = ~1 baseline SD

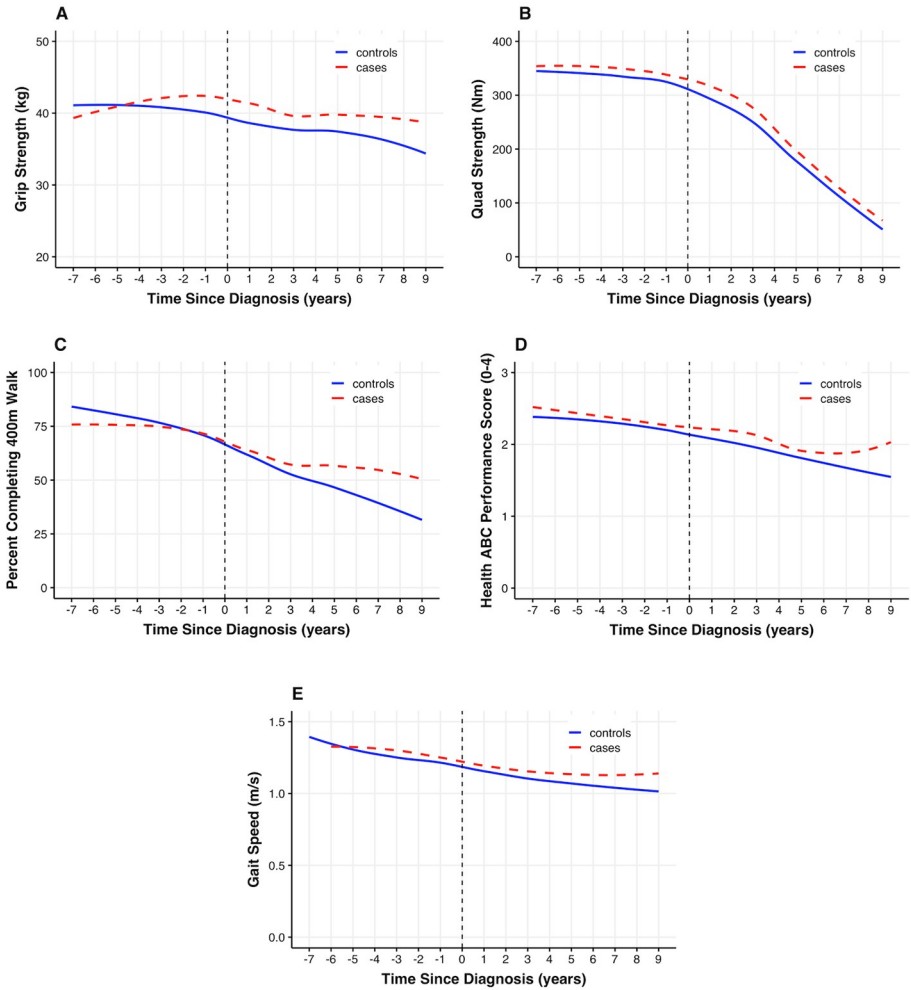

**Fig 2. Strength and physical function trajectories.** Loess regression models depicting trajectories of strength and physical function among cancer cases and controls.

the 400m walk test over time but from the point of diagnosis onwards, an increasing proportion of cases remaining in the study completed the test. In terms of the HABCPPB scores, performance declined significantly over time ($p<0.01$), though this was equivalent in cases and in controls and the rate of decline did not change after diagnosis. For 20m gait speed, both cases and controls experienced declines, with no differences in trajectory ($p = 0.89$), though cases declined at a slightly slower rate from the point of diagnosis onwards ($p<0.02$).

## Factors associated with clinically meaningful change strength and physical function

Table 3 indicates the categorization of PC cases into 4 groups (*decreasing*, *consistently low*, *consistently high*, *increasing*) representing change in upper body strength (grip strength), lower body strength (quad strength) and 20m gait speed, separately. S2–S4 Tables show the results of bivariate associations between change score groups for grip strength, quad strength, and 20m gait speed and all covariates 3–4 years post diagnosis. Men with PC who were younger at their index visit ($p = 0.03$), who had greater lean body mass ($p = 0.03$) and who had fewer depressive

symptoms ($p = 0.01$) were significantly more likely to maintain a high level of grip strength. Men with PC who were older at index visit ($p < .01$), who had more follow up years in the study ($p \leq .01$), and who were taking PC drugs ($p < 0.01$) were significantly more likely to have decreased quad strength. PC cases who had arthritis ($p = 0.02$) and who had more years of follow up in the study ($p = 0.01$) were significantly more likely to have experienced a decrease in their 20m gait speed.

## Discussion

In the current study, the trajectories of strength and physical function among both older men diagnosed with PC and among race-matched men without cancer declined at similar rates over an equivalent amount of time, with the greatest declines seen in lower leg strength as indexed by quad strength testing. Starting from approximately 80 years of age, both cases and controls lost around 70% of their quad strength over the following 6 years compared with having lost only 7% over the previous decade. Given the low rates of ADT use among this sample of men diagnosed with prostate cancer, it appears the majority of the decline in lower body strength was due to aging related changes. Unexpectedly, the overall rate of decline for grip strength was slower in men with PC compared to men without cancer, which may be a result of the fact that PC cases had significantly greater grip strength than controls at study baseline and at index visit unlike for other measures of strength and function. Previous studies examining upper and lower body strength changes among PC patients (both receiving and not receiving ADT) have found mixed results. This may relate to the specific tests being used or the time frame across which patients were tested. For example, studies using maximal leg press have found no differences in strength while those examining leg extension have found differences (Cheung). Among men with PC, mobility (proportion completing 400m walk and 20m gait speed) remained relatively intact; however, changes in muscle strength were apparent. The majority of men with PC either had significant declines in physical performance or maintained a low level of physical performance on upper and lower body strength tests (63% and 69% respectively) and gait (78%).

Our study differed from previously reported data in noteworthy ways. Alibhai and colleagues [11], examined differences in physical function between men with PC (some receiving ADT) and non-cancer controls over time and found that grip strength declined only among PC patients receiving ADT. In our study, grip strength decreased in both cases and controls, but was the most dissimilar outcome between the two groups. These estimates may have also been affected by a smaller number of participants at the beginning and end of the follow-up periods. The Alibhai study found no changes over time among any group for the timed up and go test (TUG), used as a measure of lower body strength, in contrast to our finding of a large decline in quad strength over time. The discrepancy may be due to their shorter follow up (12 months) and the slightly different testing components [23]. The most significant difference in our study was that physical function was examined before men were diagnosed with PC, illustrating the acute effects of diagnosis and first line treatments on physical function over time. To our knowledge, no other studies have used objective measures of functioning to evaluate change across this period. Moreover, we did not discriminate between men who were on/off ADT. A relatively small proportion of men in this study had ADT exposure, highlighting the significance of the change in quad strength with advancing age.

Our sample contained 50% African Americans, vs only 6% African Americans in the SEER cohort [17], a critical contribution considering that the majority of PC literature underrepresents African Americans. Reeve et al reported significant differences in physical health-related quality of life (HRQL) between cases and controls over 6 months following diagnosis,

including worse depressive symptoms. Except for grip strength, we did not find significant differences between cases and controls in objective measures of physical performance over time. Furthermore, Chambers and colleagues [24] examined trajectories of self-reported HRQL in 2250 men recently diagnosed with PC over a 6-year period, reporting poorer physical HRQL was predicted by older age, lower education, lower income, comorbidities and receiving hormonal therapy. We found no associations between level of education, number of comorbidities and any physical performance measure. Overall, among men with PC, increasing age was the factor most consistently associated with meaningful declines in function over time.

Previous studies have reported that up to 50% of men diagnosed with PC will at some point receive ADT [7]. Though use may be declining for men with low-risk early stage disease, due to its adverse effects [25], ADT remains the cornerstone of treatment for advanced PC. Preparing for possible disease progression in conjunction with inevitable aging-related functional declines may promote maintenance of physical health-related QOL among men diagnosed with early-stage disease. Largely in contrast to our hypothesis, we found men diagnosed with PC who were healthy at baseline (eligibility criteria for Health ABC study) did not experience accelerated declines in long-term physical function when compared to men without cancer. Similar self-reported changes in long-term physical function have been described among women with breast cancer [26] but other findings suggest a cancer history is associated with marked declines in short-term function [27]. In our study, all older men experienced decreases in their function over time, with the largest declines seen in lower body strength. Helping all older men, particularly those diagnosed with PC, to maintain function as they age is an important clinical goal. Previous studies show that physical function declines less rapidly during exercise intervention [28]. Therefore, it is critical to identify strategies for maintaining effective and sustainable lifestyle behaviors that support optimal physical function [29]. In addition, aerobic training to support cardiovascular health is also critical among men who are at risk of developing CVD [30, 31]. Our finding that older men are, in general, not engaging in regular exercise provides an opportunity for such interventions.

Maintaining muscle strength and quality [32] through the adoption of exercise training may be particularly important for promoting healthy survivorship among patients with PC. In the RENEW study [28], cancer survivors (breast, prostate and colorectal), in a wait-listed control group experienced significant declines in self-reported physical function over 12 months when compared with those receiving a home-based exercise intervention. In a separate study, Klepin et al. [14] found that lower extremity physical performance predicted both 2-year progression to disability and overall survival among cancer survivors from the Health ABC cohort. Therefore, early intervention to offset functional declines in aging patients with PC may be critical, with diagnosis providing an important opportunity (teachable moment) [33] to engage with these men.

We believe this study had a number of important strengths including: 1) the diverse sample (50% African Americans); with 2) multiple years of follow-up; which allowed us to 3) characterize change in physical function from pre-diagnosis; using 4) objective measures of function on multiple domains; and 5) a matched non-cancer cohort. We also recognize the following limitations: We did not have reliable information on the disease stage at diagnosis, making comparison by stage within and across other studies impossible. Relatively few men were on ADT, suggesting the majority had early stage disease at diagnosis [34] and making it unlikely that we could detect significant differences between groups. A further limitation of the study is the lack of other treatment information such as surgery or radiation as these may also have impacted aspects of strength or physical function, especially in the early post diagnosis period. Men with high risk disease or more advanced cancer at diagnosis would also likely have different treatments and/or psychosocial reactions to diagnosis. Additionally, while we did not find

specific associations between BMI and trajectories of physical function over time, the use of more precise measures of body composition may have provided better information. Finally, we did not evaluate diet or nutrition, a critical factor contributing to changes in body composition in men who begin hormone therapy [35].

In conclusion, our findings reinforce the importance of health status at diagnosis. Furthermore, as part of multidisciplinary care, all aging men should undergo assessments of strength and physical function to identify those with deficits. Routine assessments will provide a basis for the prescription of exercise programs and for assessing the risk of disablement and mortality as these men age. Older patients with PC should be prescribed resistance exercise training to promote the maintenance of muscular strength [36, 37].

## Supporting information

**S1 Table. Characteristics of the study sample at last pre-diagnosis (index) visit.**
(DOCX)

**S2 Table. Change in grip strength at 3–4 year follow-up visit.**
(DOCX)

**S3 Table. Change in quad strength at 3–4 year follow-up visit.**
(DOCX)

**S4 Table. Change in 20 m walking speed at 3–4 year follow-up visit.**
(DOCX)

## Author Contributions

**Conceptualization:** Alexander R. Lucas, Rhonda L. Bitting, Jason Fanning, W. Jack Rejeski, Heidi D. Klepin.

**Data curation:** Stephen B. Kritchevsky.

**Formal analysis:** Alexander R. Lucas, Scott Isom.

**Methodology:** Alexander R. Lucas, Rhonda L. Bitting, Jason Fanning, Scott Isom, W. Jack Rejeski, Heidi D. Klepin.

**Project administration:** Alexander R. Lucas.

**Supervision:** Stephen B. Kritchevsky.

**Writing – original draft:** Alexander R. Lucas, Rhonda L. Bitting, Jason Fanning, W. Jack Rejeski, Heidi D. Klepin.

**Writing – review & editing:** Alexander R. Lucas, Rhonda L. Bitting, Jason Fanning, Scott Isom, W. Jack Rejeski, Heidi D. Klepin, Stephen B. Kritchevsky.

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
