## [Decision Letter · Decision Letter 0]

7 Nov 2019

PONE-D-19-28019

Trajectories in Muscular Strength and Physical Function Among Men with and without Prostate Cancer in the Health Aging and Body Composition Study

PLOS ONE

Dear Dr. Lucas,

Thank you for submitting your manuscript to PLOS ONE. After careful consideration, we feel that it has merit but does not fully meet PLOS ONE’s publication criteria as it currently stands. Therefore, we invite you to submit a revised version of the manuscript that addresses the points raised during the review process.

We would appreciate receiving your revised manuscript by Dec 22 2019 11:59PM. To enhance the reproducibility of your results, we recommend that if applicable you deposit your laboratory protocols in protocols.io, where a protocol can be assigned its own identifier (DOI) such that it can be cited independently in the future. For instructions see: http://journals.plos.org/plosone/s/submission-guidelines#loc-laboratory-protocols

We look forward to receiving your revised manuscript.

Kind regards,

Justin C. Brown

Academic Editor

PLOS ONE

Journal Requirements:

"The institutional review boards of the study sites approved the protocol."

"Dr. Lucas’s work on this project was partly supported by a National Cancer Institute training grant (R25 CA122061). Dr. Klepin receives support from Wake Forest University Claude D. Pepper Older Americans Independence Center (P30-AG21332). Dr. Isom is supported by National Cancer Institute’s Cancer Center Support Grant award number P30CA012197 issued to the Wake Forest Baptist Comprehensive Cancer Center. This research was supported by National Institute on Aging (NIA) Contracts N01-AG-6-2101; N01-AG-6-2103; N01-AG-6-2106; NIA grant R01-AG028050, and NINR grant R01-NR012459. This research was funded in part by the Intramural Research Program of the NIH, National Institute on Aging.The funders had no role in study design, data collection and analysis, decision to publish, or preparation of the manuscript.".

i) Please provide an amended statement that declares *all* the funding or sources of support (whether external or internal to your organization) received during this study, as detailed online in our guide for authors at http://journals.plos.org/plosone/s/submit-now.  Please also include the statement “There was no additional external funding received for this study.” in your updated Funding Statement.

ii) Please include your amended Funding Statement within your cover letter. We will change the online submission form on your behalf.

Reviewers' comments:

Reviewer's Responses to Questions

**Comments to the Author**

1. Is the manuscript technically sound, and do the data support the conclusions?

Reviewer #1: Partly

Reviewer #2: Yes

2. Has the statistical analysis been performed appropriately and rigorously? 

Reviewer #1: Yes

Reviewer #2: Yes

3. Have the authors made all data underlying the findings in their manuscript fully available?

Reviewer #1: Yes

Reviewer #2: Yes

4. Is the manuscript presented in an intelligible fashion and written in standard English?

Reviewer #1: Yes

Reviewer #2: Yes

5. Review Comments to the Author

Reviewer #1: Thank you for the opportunity to review this manuscript. The purpose of this research was to compare longitudinal changes in strength and physical function between prostate cancer survivors and matched controls in a cohort of community living older adults. A large body of literature supports muscle mass and muscle strength loss after treatment for prostate cancer; however, very few studies have captured objective assessments of strength and function prior to diagnosis. This research provides valuable insight into trajectories of change in these facets of aging both prior to and after a prostate cancer diagnosis among older men. Comments and suggestions are detailed below.

Introduction:

•The introduction of this study does a sufficient job introducing the rationale behind the strength and physical function concerns with prostate cancer; however, it focuses on the specific effects of ADT. This does not seem to carry through the rest of the paper in terms of analyses. It would be beneficial for this section to expand to include the effects of prostate cancer more broadly, rather than those of ADT.

•It’s unclear how depressive symptoms are associated with strength and physical function for inclusion as a covariate. Please elaborate on the effects of depressive symptoms on strength and physical function in this population.

Methods:

•The authors may consider separately defining muscle strength and physical function. The methods section would be strengthened by specifically describing how each construct was operationalized.

•The authors may consider defining self-reported physical function in the previous section “muscular strength and physical function” with the other measures of physical function rather than with the covariates.

•It is surprising that a cohort study focused on body composition would only include body mass index in an analysis of muscle strength and physical function. It seems that muscle mass would be an important factor in this relationship, and it is unclear why this would be omitted.

•The author may consider including details on which measures were continuous or categorical for analysis.

•Please include methodology for “placement” into decreasing, consistently low, consistently high, and increasing group.

•Please clarify the use of baseline versus index visit results. The authors may consider including more detail on the index visit (e.g. time since baseline visit).

Results:

•Please provide data on the number of participants at each visit included in this analysis.

•Please include the HABCPPB acronym definition in the notes for Table 1.

•If available, please provide descriptive data on prostate cancer diagnoses (Gleason score, treatment type).

Discussion:

•Please provide further rationale for cases’ preservation of upper body strength as compared to controls. (paragraph 1, page 13)

•Please consider discussing the sensitivity of measurement techniques used to assess lower body strength. Many studies have found differences in upper body strength, but not lower body. The authors may consider referencing Cheung, 2014. (paragraph 2, page 13)

•It is unclear if the observed declines in function have more to do with age-related decline than with cancer diagnosis. Please elaborate. (paragraph 1, page 14)

•The authors state that resistance exercise should be utilized to maintain muscular strength in those treated with ADT. Given that the authors stated the proportion of patients treated with ADT appeared lower than previously reported prevalence rates, this conclusion extends beyond this scope of these results. Please consider revising this statement. (paragraph 3, page 15)

•The authors include body mass index as a limitation of this study; however, other publications from this cohort include DXA. It is unclear why the authors chose to not include these measurements and instead used an assessment that is known to be a limitation to this study. (paragraph 2, page 15)

•The statement regarding older prostate cancer patients contradicts the previous statement that older patients did not fare worse than younger patients provided they were healthy at baseline. Please revise. (paragraph 2, page 14; paragraph 3, page 15)

Reviewer #2: The present study by Lucas et al describes the trajectories in muscular strength and physical function in older men with prostate cancer compared to a matched sample of non-cancer controls from within the Health ABC dataset. The study utilizes a unique data source (the HABC cohort study) to examine differences in strength and function both before and after a cancer diagnosis, and utilizes several objective measures. The study ultimately finds significant age-related declines in measures of strength and function across both groups, but no difference in the prostate cancer cases compared to non-cancer controls. The manuscript is well written and the study is well designed. Overall, this is an interesting study and presents novel data on these measures in older adults with prostate cancer. Only a few comments for the authors.

- How exactly were the non-cancer cases chosen? The methods suggest the frequency matching was weighted by race. Also the discussion says it was age- and race-matched, but I find this is not well explained within the methods and is critically important to the study.

- The lack of cancer stage is a significant limitation. I believe within the HABC dataset there is some information regarding limited vs. metastatic disease, and may want to include this information at least in the patient characteristics table.

- I would also add the androgen deprivation information into table 1 as part of the description of the cohort. How was the use of ADT attained? May be helpful to look at the overall % receiving ADT as part of prostate cancer therapy, rather than just having the range in proportions for any given year. I realize this sub-population may have been too small to examine, but any analyses focused on this group would be of great interest (which I believe the authors well-recognize).

- Why do the authors think there were baseline differences in HABCPPB, grip strength, and quad strength between cases and controls? May be worth further elaborating in the discussion.

- As there are several other therapies utilized in prostate cancer treatment other than androgen deprivation, I would be sure to explain the lack of treatment information as a limitation within the discussion.

Minor comments

- Typically manuscripts are written in the past tense “i.e. ‘The aims of our study were…’ etc.”, but I leave that to the discretion of the authors/journal staff as otherwise the manuscript is well written.

- On page 15, line 74, there is a stray 2 after the word found

6. PLOS authors have the option to publish the peer review history of their article (what does this mean?). If published, this will include your full peer review and any attached files.

Reviewer #1: No

Reviewer #2: No

---

## [Author Response · Author response to Decision Letter 0]

20 Jan 2020

January 20th, 2020

Justin C. Brown, PhD

Dear Dr. Brown and Reviewers:

We thank you for the positive feedback on our manuscript and we appreciate the opportunity to respond to and address the areas of concern. Our point-by-point responses and details of changes are below. We have highlighted the changes in the revised manuscript.

Journal Requirements:

Response: Thank you for the guidance, we believe our manuscript complies with your style requirements.

"The institutional review boards of the study sites approved the protocol."

Response: We have now added the more detailed statement to include the full name of the IRB’s that approved our study. This is in line with how the HABC Study ethics statements have been described in other recently published PLOS ONE manuscripts.

Text in the methods now reads: “The Institutional Review Boards at the University of Pittsburgh, the University of Tennessee and the University of California, San Francisco approved the Health ABC protocol.”

Response: Done

"Dr. Lucas’s work on this project was partly supported by a National Cancer Institute training grant (R25 CA122061). Dr. Klepin receives support from Wake Forest University Claude D. Pepper Older Americans Independence Center (P30-AG21332). Dr. Isom is supported by National Cancer Institute’s Cancer Center Support Grant award number P30CA012197 issued to the Wake Forest Baptist Comprehensive Cancer Center. This research was supported by National Institute on Aging (NIA) Contracts N01-AG-6-2101; N01-AG-6-2103; N01-AG-6-2106; NIA grant R01-AG028050, and NINR grant R01-NR012459. This research was funded in part by the Intramural Research Program of the NIH, National Institute on Aging. The funders had no role in study design, data collection and analysis, decision to publish, or preparation of the manuscript.".

i) Please provide an amended statement that declares *all* the funding or sources of support (whether external or internal to your organization) received during this study, as detailed online in our guide for authors at http://journals.plos.org/plosone/s/submit-now. Please also include the statement “There was no additional external funding received for this study.” in your updated Funding Statement.

ii) Please include your amended Funding Statement within your cover letter. We will change the online submission form on your behalf.

Response: We have updated the funding statement to include all funding sources of support during the study.

Reviewers' comments:

Reviewer's Responses to Questions

Comments to the Author

1. Is the manuscript technically sound, and do the data support the conclusions?

Reviewer #1: Partly

Reviewer #2: Yes

2. Has the statistical analysis been performed appropriately and rigorously? 

Reviewer #1: Yes

Reviewer #2: Yes

3. Have the authors made all data underlying the findings in their manuscript fully available?

Reviewer #1: Yes

Reviewer #2: Yes

4. Is the manuscript presented in an intelligible fashion and written in standard English?

Reviewer #1: Yes

Reviewer #2: Yes

5. Review Comments to the Author

Reviewer #1: Thank you for the opportunity to review this manuscript. The purpose of this research was to compare longitudinal changes in strength and physical function between prostate cancer survivors and matched controls in a cohort of community living older adults. A large body of literature supports muscle mass and muscle strength loss after treatment for prostate cancer; however, very few studies have captured objective assessments of strength and function prior to diagnosis. This research provides valuable insight into trajectories of change in these facets of aging both prior to and after a prostate cancer diagnosis among older men. Comments and suggestions are detailed below.

Introduction: 

•The introduction of this study does a sufficient job introducing the rationale behind the strength and physical function concerns with prostate cancer; however, it focuses on the specific effects of ADT. This does not seem to carry through the rest of the paper in terms of analyses. It would be beneficial for this section to expand to include the effects of prostate cancer more broadly, rather than those of ADT.

Response: We agree with the reviewer and have now expanded the introduction and rationale for the study, to include more than just a description of the specific effects of ADT. As part of this expansion we have included information relative to the point made below regarding the effects of depressive symptoms on strength and physical function.

•It’s unclear how depressive symptoms are associated with strength and physical function for inclusion as a covariate. Please elaborate on the effects of depressive symptoms on strength and physical function in this population.

Response: A recent study has examined the relationship between depressive symptoms and grip strength in the NHANES data for adults over 60 years of age (Brooks, et al. 2018). Findings showed that increasing levels of depressive symptoms were associated with diminished grip strength. Other recent studies have found relationships between depression and physical functioning among older adults in nursing homes (Kvael, et al. 2017) and in men and women with and without Alzheimer’s disease (Watts, et al. 2018). The rationale for these linkages is that older adults who are depressed may tend to limit their engagement in previously important activities that may further result in declining functional status. We have added this to the introduction (page 1, paragraph 1).

Methods: 

•The authors may consider separately defining muscle strength and physical function. The methods section would be strengthened by specifically describing how each construct was operationalized.

Response: We have separately operationalized muscle strength and physical function constructs. 

•The authors may consider defining self-reported physical function in the previous section “muscular strength and physical function” with the other measures of physical function rather than with the covariates.

Response: We have decided to keep the self-reported measures of functional status under the section on covariates. This decision is based on the rationale that we are separating the influence of perceived capacity to engage in these tasks from the actual measures of strength and physical function.

•It is surprising that a cohort study focused on body composition would only include body mass index in an analysis of muscle strength and physical function. It seems that muscle mass would be an important factor in this relationship, and it is unclear why this would be omitted.

Response: The reviewer is correct. The Health ABC study did include more precise measures of body composition. CT scans were only performed at years 1, 6 and 10 and 11; however, DXA scans for assessing body fat % and lean body mass did occur at years 1-6, 8 and 10, therefore we have now included body fat % and lean body mass from those scans where data were not missing. These data are now included in Table 1, and S1-S4 Tables.

•The author may consider including details on which measures were continuous or categorical for analysis.

Response: The requested information on variables is included in all tables. Variables shown as Yes or No are categorical while those with a mean (SD) after the variable name are continuous variables.

•Please include methodology for “placement” into decreasing, consistently low, consistently high, and increasing group.

Response: On page 7, paragraph 1, we have expanded upon the description of how cases were placed into groups.

The relevant text now reads:

“To assess the association of relevant covariates with trajectories of upper body strength (grip strength), lower body strength (quad strength) and gait (20m usual pace gait speed), we placed PC cases into 1 of 4 groups for each outcome as follows: group 1 - decreasing; group 2 – consistently low; group 3 – consistently high; group 4 - increasing. Cut offs for determining a meaningful change (±) and the groups men were placed into (decreasing, consistently low, consistently high or increasing) are shown in Table 3 and were based on currently published literature, except in the case of quad strength (a 1.0 standard deviation (SD) was used).”

•Please clarify the use of baseline versus index visit results. The authors may consider including more detail on the index visit (e.g. time since baseline visit).

Response: First, we presented baseline characteristics to examine the difference or similarity of cases to control prior to prostate diagnosis. Second, under Results in paragraph 1, page 10 we have added text to describe the Table S1 showing patient characteristics at the index visit. We have also added the following text. 

“S1 Table shows sample characteristics at index visit (last visit before diagnosis, including a breakdown of the proportion of patients and matched controls who had their index visit in years 1 (baseline), 2, 4, 6 and 8, which ranged from 9% in year 8 to 33.35% in year 2.” 

Results:

•Please provide data on the number of participants at each visit included in this analysis.

Response: As mentioned under the previous point. S1 Table shows the breakdown of identified cases and matched controls by study visit/year. 

•Please include the HABCPPB acronym definition in the notes for Table 1.

Response: Done

•If available, please provide descriptive data on prostate cancer diagnoses (Gleason score, treatment type).

Response: Unfortunately the HABC study was not focused on cancer outcomes, therefore limited clinical data for the purposes of describing prostate cancer diagnoses exist. Exceptions were adjudication of cancer diagnoses and the medications list, which we used to identify ADT.

Discussion:

•Please provide further rationale for cases’ preservation of upper body strength as compared to controls. (paragraph 1, page 13)

Response: We have provided further rationale for the preservation of upper body strength. The text in paragraph 1, page 14 now reads:

“Unexpectedly, the overall rate of decline for grip strength was slower in men with PC compared to men without cancer, which may be a result of the fact that PC cases had significantly greater grip strength than controls at study baseline and at index visit unlike for other measures of strength and function.”

•Please consider discussing the sensitivity of measurement techniques used to assess lower body strength. Many studies have found differences in upper body strength, but not lower body. The authors may consider referencing Cheung, 2014. (paragraph 2, page 13)

Response: We have now added a brief discussion regarding sensitivity of measurement. The text reads:

“Previous studies examining upper and lower body strength changes among PC patients (both receiving and not receiving ADT) have found mixed results. This may relate to the specific tests being used or the time frame across which patients were tested. For example, studies using maximal leg press have found no differences in strength while those examining leg extension have found differences (Cheung, 2014).”

•It is unclear if the observed declines in function have more to do with age-related decline than with cancer diagnosis. Please elaborate. (paragraph 1, page 14)

Response: We believe in this cohort of well-functioning older men, that the observed declines are predominantly due to age-related changes and not to cancer diagnosis per se. This has been added on page 14 paragraph 1.

•The authors state that resistance exercise should be utilized to maintain muscular strength in those treated with ADT. Given that the authors stated the proportion of patients treated with ADT appeared lower than previously reported prevalence rates, this conclusion extends beyond this scope of these results. Please consider revising this statement. (paragraph 3, page 15)

Response: We have removed the above statement from our conclusions based on the limited data we have presented.

•The authors include body mass index as a limitation of this study; however, other publications from this cohort include DXA. It is unclear why the authors chose to not include these measurements and instead used an assessment that is known to be a limitation to this study. (paragraph 2, page 15)

Response: We have now included all DXA measures of body fat % and lean body mass where possible and removed the statement of BMI as a limitation.

•The statement regarding older prostate cancer patients contradicts the previous statement that older patients did not fare worse than younger patients provided they were healthy at baseline. Please revise. (paragraph 2, page 14; paragraph 3, page 15)

Response: we have revised the above referenced statements to reflect that: first, among men with PC, increasing age was the factor most associated with declines in function; second, we clarified that in contrast to our hypotheses, men with PC did not experience accelerated declines in physical function compared to men without cancer.

Reviewer #2: The present study by Lucas et al describes the trajectories in muscular strength and physical function in older men with prostate cancer compared to a matched sample of non-cancer controls from within the Health ABC dataset. The study utilizes a unique data source (the HABC cohort study) to examine differences in strength and function both before and after a cancer diagnosis, and utilizes several objective measures. The study ultimately finds significant age-related declines in measures of strength and function across both groups, but no difference in the prostate cancer cases compared to non-cancer controls. The manuscript is well written and the study is well designed. Overall, this is an interesting study and presents novel data on these measures in older adults with prostate cancer. Only a few comments for the authors.

- How exactly were the non-cancer cases chosen? The methods suggest the frequency matching was weighted by race. Also the discussion says it was age- and race-matched, but I find this is not well explained within the methods and is critically important to the study.

Response: Thank you for catching this discrepancy. The description in the methods is accurate and the cases and controls were identified by frequency matching, weighted by race, to randomly assign an index visit for the non-cancer controls at a ratio of 4:1 for a total analytic sample of 585 men. We have removed the text in the discussion referring to age-matched.

- The lack of cancer stage is a significant limitation. I believe within the HABC dataset there is some information regarding limited vs. metastatic disease, and may want to include this information at least in the patient characteristics table.

Response: We were unable to identify any information regarding cancer stage, including the limited vs metastatic disease. Given the low number of patients in our cohort with ADT (4 patients) we believe most were likely early stage disease, but we cannot verify that.

- I would also add the androgen deprivation information into table 1 as part of the description of the cohort. How was the use of ADT attained? May be helpful to look at the overall % receiving ADT as part of prostate cancer therapy, rather than just having the range in proportions for any given year. I realize this sub-population may have been too small to examine, but any analyses focused on this group would be of great interest (which I believe the authors well-recognize).

Response: 

Over the course of all the visits there were 17 instances of subjects using 4 ADT drugs (based on the codes for the medications list of the study: Leuprolide, Goserelin, Buserelin, Nafarelin). These instances were for 12 unique people and only 4 of these people are included in our dataset (all 4 are in the cancer group). These 4 had an index year prior to the drug appearing in their records (i.e. index year = 2 and medication found in year 3). Each only has one visit (1 year post index visit for 3 participants and 2 years post for the other) with the drug in their record with all having at least 1 visit after the drug was recorded where it was no longer one of their listed medications. Due the small numbers we are unfortunately unable to conduct any further analyses using ADT as a sub-population. This has been added to page 10, paragraph 1.

- Why do the authors think there were baseline differences in HABCPPB, grip strength, and quad strength between cases and controls? May be worth further elaborating in the discussion.

Response: One possibility as to why the cases and controls may have differed in baseline measures of grip strength, quad strength and HABCPPB may be the relatively smaller number of PC cases compared with controls, reflected by less variability in the control group. While these differences were significant, they no longer existed by the index visit and the difference was still within the cut off for meaningful change, as used to place men into groups (decreasing, consistently low, consistently high or increasing).

- As there are several other therapies utilized in prostate cancer treatment other than androgen deprivation, I would be sure to explain the lack of treatment information as a limitation within the discussion.

Response: We have added the lack of treatment information as a limitation. (page 16, paragraph 3)

Minor comments

- Typically manuscripts are written in the past tense “i.e. ‘The aims of our study were…’ etc.”, but I leave that to the discretion of the authors/journal staff as otherwise the manuscript is well written.

Response: We have corrected instances of incorrect tense.

- On page 15, line 74, there is a stray 2 after the word found

Response: removed

---

## [Editor Report · Decision Letter 1]

24 Jan 2020

Trajectories in Muscular Strength and Physical Function Among Men with and without Prostate Cancer in the Health Aging and Body Composition Study

PONE-D-19-28019R1

Dear Dr. Lucas,

We are pleased to inform you that your manuscript has been judged scientifically suitable for publication and will be formally accepted for publication once it complies with all outstanding technical requirements.

With kind regards,

Justin C. Brown

Academic Editor

PLOS ONE
---

## [Editor Report · Acceptance letter]

31 Jan 2020

PONE-D-19-28019R1 

Trajectories in Muscular Strength and Physical Function Among Men with and without Prostate Cancer in the Health Aging and Body Composition Study 

Dear Dr. Lucas:

I am pleased to inform you that your manuscript has been deemed suitable for publication in PLOS ONE. Congratulations! Your manuscript is now with our production department. 

With kind regards,

on behalf of

Dr. Justin C. Brown 

Academic Editor

PLOS ONE